# A region-resolved mucosa proteome of the human stomach

Xiaotian Ni[1,2,3], Zhaoli Tan[1], Chen Ding[2,4], Chunchao Zhang[5], Lan Song[2,6], Shuai Yang[1], Mingwei Liu[2], Ru Jia[1], Chuanhua Zhao[1], Lei Song[2], Wanlin Liu[2], Quan Zhou[2], Tongqing Gong[2], Xianju Li[2], Yanhong Tai[1], Weimin Zhu[2], Tieliu Shi[3], Yi Wang[2,5], Jianming Xu[1], Bei Zhen[2] & Jun Qin[2,4,5]

The human gastric mucosa is the most active layer of the stomach wall, involved in food digestion, metabolic processes and gastric carcinogenesis. Anatomically, the human stomach is divided into seven regions, but the protein basis for cellular specialization is not well understood. Here we present a global analysis of protein profiles of 82 apparently normal mucosa samples obtained from living individuals by endoscopic stomach biopsy. We identify 6,258 high-confidence proteins and estimate the ranges of protein expression in the seven stomach regions, presenting a region-resolved proteome reference map of the near normal, human stomach. Furthermore, we measure mucosa protein profiles of tumor and tumor nearby tissues (TNT) from 58 gastric cancer patients, enabling comparisons between tumor, TNT, and normal tissue. These datasets provide a rich resource for the gastrointestinal tract research community to investigate the molecular basis for region-specific functions in mucosa physiology and pathology including gastric cancer.

[1] Department of Gastrointestinal Oncology, The Fifth Medical Center, General Hospital of PLA, Beijing 100071, China. [2] State Key Laboratory of Proteomics, Beijing Proteome Research Center, National Center for Protein Sciences (The PHOENIX Center, Beijing), Institute of lifeomics, Beijing 102206, China. [3] Center for Bioinformatics, East China Normal University, Shanghai 200241, China. [4] State Key Laboratory of Genetic Engineering, Human Phenome Institute, Institutes of Biomedical Sciences, School of Life Sciences, Zhongshan Hospital, Fudan University, Shanghai 200032, China. [5] Alek Center for Molecular Discovery, Verna and Marrs McLean Department of Biochemistry and Molecular Biology, Department of Molecular and Cellular Biology, Baylor College of Medicine, Houston, TX 77030, USA. [6] Department of Bioinformatics, College of Life Science, Hebei University, Baoding 071002, China. These authors contributed equally: Xiaotian Ni, Zhaoli Tan, Chen Ding. Correspondence and requests for materials should be addressed to J.X. (email: jmxu2003@yahoo.com) or to B.Z. (email: zp1963@sina.com) or to J.Q. (email: jqin@bcm.edu)

The availability of a human proteome reference map with direct measurements of peptides and proteins can enable the translation of basic research into clinical practice. Recently, a composite draft map of the human proteome has systematically identified and annotated protein-coding genes with proteomic profiling of more than 30 histologically normal human tissues. Moreover, proteogenomic profiling of several human cancers, including colorectal, ovarian, and breast cancer as well as a proteomic profiling of the diffuse-type gastric cancer (GCA) have also been described[1–4]. As molecular effectors of cellular functions that are directly involved in all biological processes, proteins provide the most valuable information and insights for our understanding of signaling pathways in physiology and pathology.

Alterations of many properties of cancers can be derived by comparing the proteome of tumor tissues with the regions surrounding tumors (a.k.a. tumor nearby tissue, TNT) from the same patient. TNT appears to be histologically normal and is often presumed to be functionally normal. However, it is still debatable whether TNT is truly normal as it comes from cancer patients. Ideally, truly normal tissues (N) should come from nondiseased individuals, but they are difficult to obtain due to ethical reasons. Apparently normal tissues from autopsy samples, which have been used in several previous studies including the draft proteome and the human proteome atlas[5,6] may not be considered truly normal as the samples came from nonliving individuals.

A true human proteome reference map also needs to take into the consideration of individual differences (i.e., interpersonal heterogeneity). While it is not possible to obtain T, TNT, and N tissues from the same person, a collection of the N tissues obtained from a sufficient number of normal individuals could in principle cover the interpersonal heterogeneity. In clinical diagnostics such as blood and urine tests, this issue is often circumvented by comparing to a reference range that is generated and covers the whole range of the variation from the population. However, no proteome reference map of human organ in physiological conditions is hitherto available.

Healthy tissues can only be obtained from limited tissue types, such as the breast tissue, where healthy tissue samples can be readily obtained from reduction mammoplasty and prophylactic mastectomy[7]. Another possible tissue source is the endoscopic biopsy of stomach from living individuals. The stomach is a roughly crescent-shaped organ in the gastrointestinal tract. It functions mainly in breaking down and digesting chewed food for further digestion and absorption in intestine[8]. The wall of the stomach is made of four layers, namely, mucosa, submucosa, muscularis externa, and the serosa. Mucosa is the innermost layer that plays roles in secretion, digestion, and absorption[9–12]. It consists mainly of the gastric glands for secretion, and is protected by the mucosa barrier including a sticky neutralizing mucus coat from self-digestion in the corrosive acidic environments[13–19].

GCA is the third most common cause for cancer-related death in the world, and more than 90% of GCA develops from the mucosa layer of the stomach[20–25]. Dissection of the molecular profiles of the mucosa from the normal stomach, the GCA tissue and its adjacent, apparently normal tissue will facilitate better understanding of GCA development and the discovery of potential markers in the early stages of GCA.

Anatomically, the human stomach is divided into seven regions known as cardia (Ca), fundus (Fu), lesser curvature (LC), greater curvature (GC), angular incisures (AI), antrum (An), and pylorus (Py)[26,27]. The division of labor and collaboration of these seven parts of the stomach is delicate and precise[28]. The gastric juice secretion, mucous membrane formation, stomach contraction, and nutrients absorption are processed orderly. For example, the stomach can be divided into acid-secreting and nonacid-secreting regions, and the acid-secreting gastric mucosa is populated in the corpus and fundus regions[29,30]. How protein expression patterns differ and what the molecular basis is for different functions in different regions is not well understood.

To map the mucosa proteome of human stomach and to determine the basis for cellular specialization in the mucosa layer, we perform a global analysis of protein profiles of 82 apparently normal mucosa samples from the seven regions obtained from stomach endoscopic biopsy. We apply the fast-seq technique[31] of liquid chromatography (LC) coupled with high-resolution mass spectrometry (MS), and identify 6258 high-confidence proteins for the human mucosa proteome with anatomical resolution of the seven stomach regions, providing a human tissue-based proteome reference map obtained from living individuals. We also measure tumor and TNT mucosa from 58 GCA patients, enabling comparisons between tumor, TNT, and normal tissues. These datasets serve as a rich resource for the gastrointestinal tract research community to investigate the molecular basis for region-specific functions in mucosa physiology and pathology including GCA.

## Results

**A region-resolved reference map of the human stomach mucosa.** As it is ethically difficult to justify the acquisition of true normal mucosa biopsy samples, we turned to clinical cases diagnosed with mild gastric ailments including superficial gastritis, bile reflux gastritis, helicobacter pylori infection, and other stomach disorders (Supplementary Data 1). To generate a reference mucosa proteome, we collected the mucosa samples from histologically normal regions, although other regions of the same stomach may have certain degree of gastric disorder. Under these conditions, we obtained at least 10 cases from each of the seven stomach regions, including Ca ($n = 12$), Fu ($n = 12$), LC ($n = 11$), GC ($n = 12$), AI ($n = 15$), An ($n = 10$), and Py ($n = 10$) (Supplementary Fig. 1a), for a total of 82 tissue samples of apparently normal stomach mucosa from 36 individuals (Supplementary Data 1). To avoid repeated sampling in the same region, only one tissue sample was taken from that region from the same person.

For in-depth mapping of the mucosa proteome, we performed LC–tandem MS (LC–MS/MS) in a high-resolution quadrupole Orbitrap mass spectrometer (Q-Exactive HF) (Fig. 1a)[32,33]. The performance of MS was assessed by running 26 quality control samples. The high reproducibility was indicated by the high interexperiment correlation coefficients (Spearman correlation coefficient > 0.8, Supplementary Fig. 1b). A total of 13,401 gene products (GPs) were called at 1% peptide false discovery rate (FDR) (Fig. 1b and sheet 1 in Supplementary Data 2). We further restricted for GPs that were detected with at least one unique peptide and two high-confidence peptides (mascot ion score > 20), and required that the proteins had to be detected in at least two cases from 1 of the 7 regions, resulting in 6258 gene products of high confidence (Fig. 1b and sheet 2 in Supplementary Data 2). On average, the mucosa proteome of our dataset contained 5670 GPs per region, ranging from a minimum of 5502 IDs in LC to a maximum of 5883 IDs in Py (Fig. 1b and Supplementary Fig. 2a).

We also found proteins that are enriched in each of the seven regions (sheet 4 in Supplementary Data 2). We selected the GPs that were reproducibly seen (≥75% of times) in 1 or 2 regions, but were rarely seen (≤25% of times) in 3 or more regions to define them as region-specific proteins. Under these conditions, we found that 19 GPs were Py enriched, 14 GPs were Py/An enriched (sheet 5 in Supplementary Data 2), and others were distal or proximal enriched. Interestingly, the majority of the Py/An

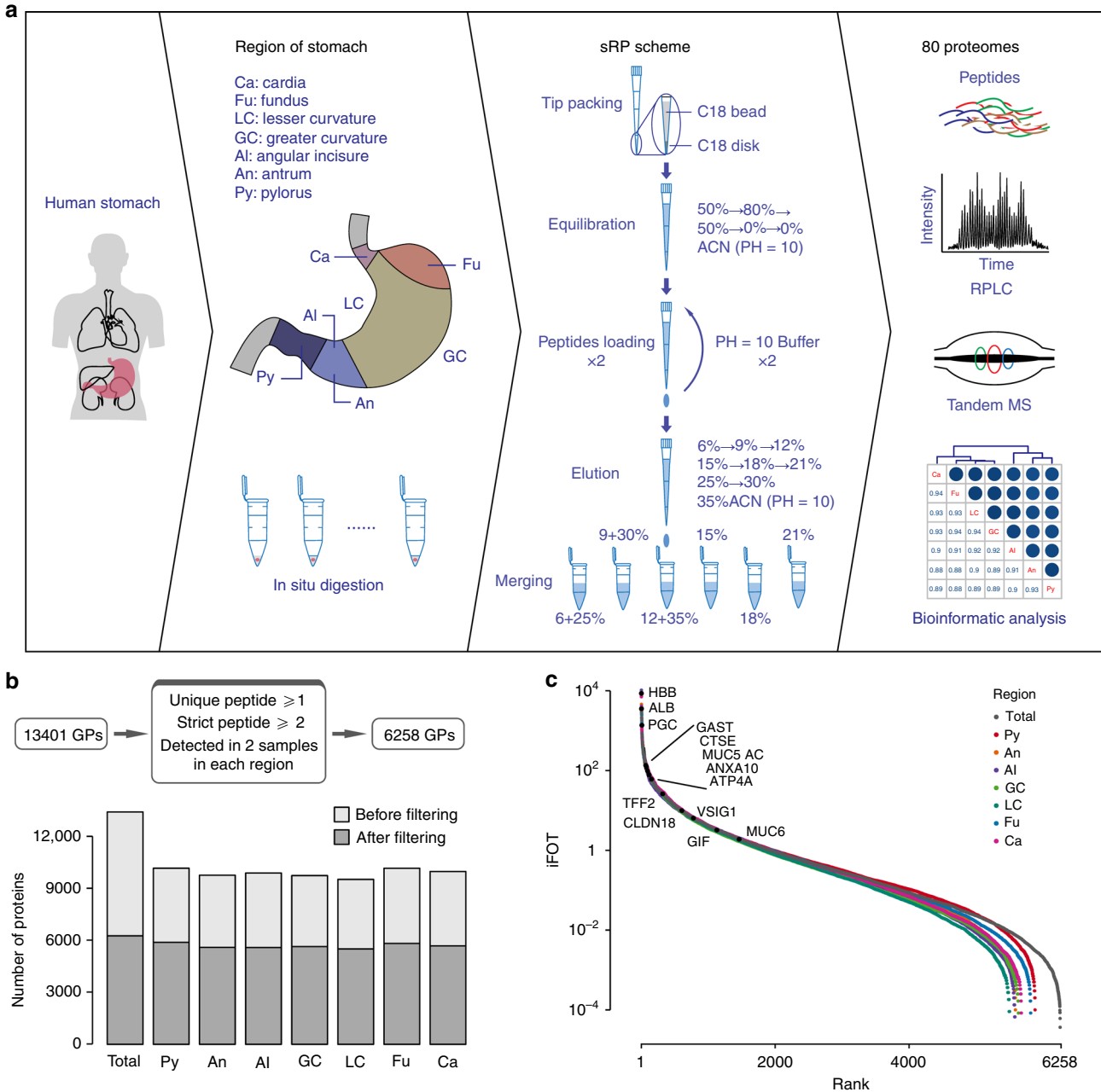

**Fig. 1** A brief summary of proteomic analysis of human gastric mucosa. **a** Illustration of sample collection, preparation, and LC–MS/MS. Human mucosa tissue samples were collected from seven antonymic stomach regions by gastro-endoscope. Mucosa tissues were in-solution digested with trypsin, and the resulting peptides were separated by sRP-RP prior to LC–MS/MS analysis. **b** A total of 13,401 GPs were identified in all 82 samples at 1% peptide-level FDR. Further screening generated 6258 high-confidence proteins that were detected with at least one unique peptide and two strict peptides in a minimum of two samples in one region. Light gray displays the number of GPs before filtering; dark gray shows the number of GPs after filtering. **c** The dynamic ranges of mucosa proteomes and several high- and low-abundant gastric proteins

enriched GPs (TRIM22, RNASEH2A, GATA6, SCAF11, EXOSC3, POLD2, and KPNA2) mainly function in RNA/DNA metabolic processes (sheet 6 in Supplementary Data 2).

The protein abundance was first calculated by intensity-based absolute quantification (iBAQ), then normalized as fraction of total (FOT), allowing for comparison among different experiments (Supplementary Fig. 1c). Within the same region, the proteomes were similar as evidenced by the large Spearman correlation coefficients (≥0.7) between the samples (Supplementary Fig. 2b). In addition, the reference proteome is highly dynamic, spanning about seven orders of magnitudes measured by the protein abundances (iFOT) (Fig. 1c). The top-ranked

proteins in the mucosa include HBB, ALB, and stomach cell markers (PGC, MUC5AC, GAST, GIF, and ATP4A). This dataset provided an in-depth human stomach mucosa proteome reference map in near physiological conditions (Supplementary Fig. 2c).

**The human stomach mucosa can be classified into two sections.** We provided a bird's-eye view of global protein expressions in a circular map to address the similarities and diversities among the region-specific proteomes (Fig. 2a). We found that 75.8% of the high-confidence GPs (4742/6258) are ubiquitously seen in all the

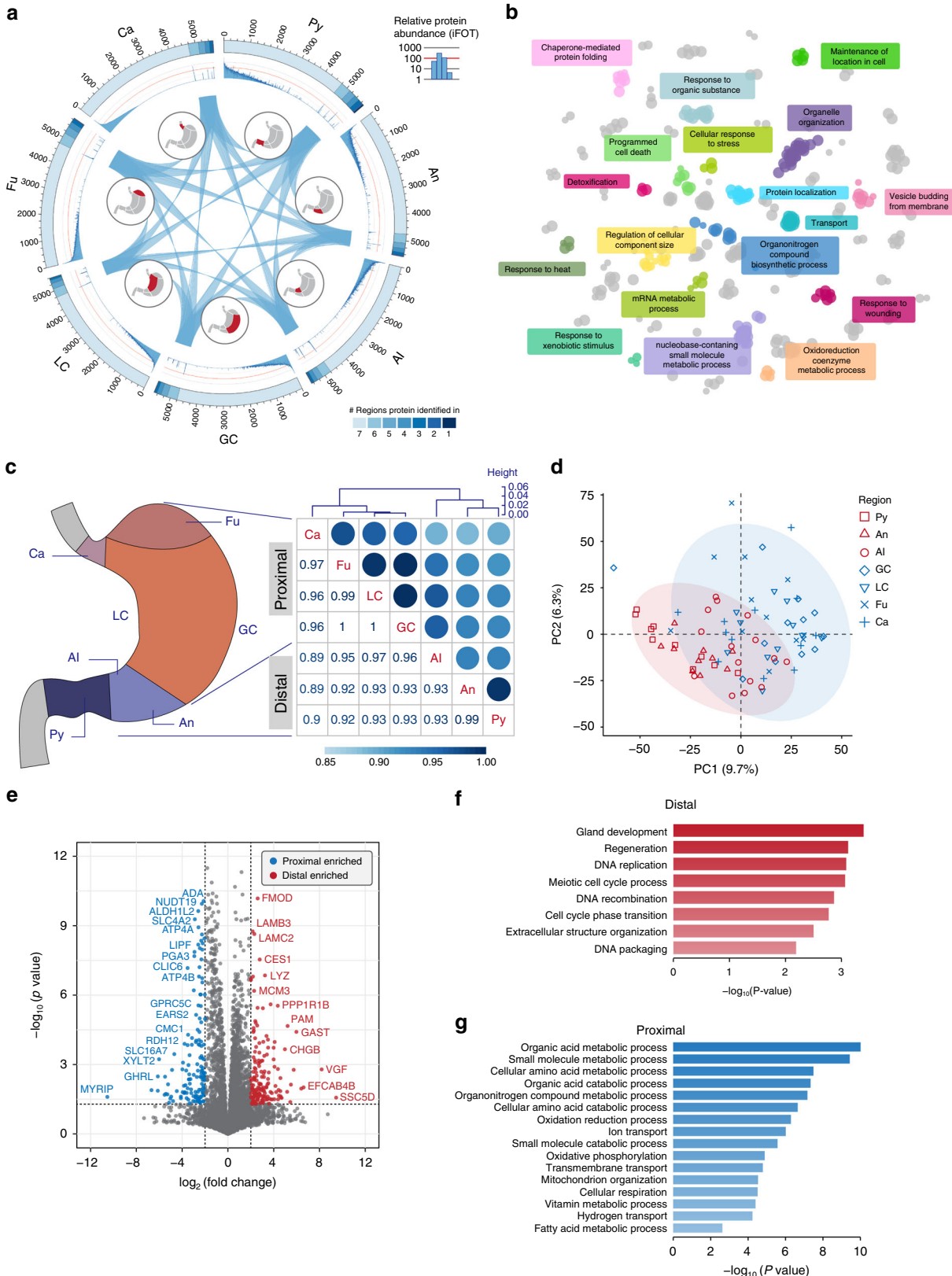

seven regions, which we designate them as the mucosa core proteome (Supplementary Fig. 3a). They are highly expressed (Fig. 2a, blue histogram) and significantly enriched in biological functions including metabolic process, localization, transport, response to organic substance, among others (Fig. 2b, Supplementary Fig. 3b, sheet 1 in Supplementary Data 3).

By correlating proteins across the regions, we found that the mucosa proteomes of the seven regions can be grouped into two substomach sections: the proximal section that includes Ca, Fu, LC, and GC, and the distal section that is composed of AI, An, and Py (Fig. 2c). The same classification was also supported by the principal component analysis (PCA) (Fig. 2d). Our finding

**Fig. 2** Spatial diversity of the human mucosa proteomes. **a** A circular proteome map displays the similarities and differences of the mucosa proteomes from seven regions. Color scales show the number of protein IDs in each region. The lightest blue color represents the core proteome that were seen in all seven regions. The relative abundance of each protein within a given region is represented by a blue histogram. **b** GO-term enrichment analysis of the core proteome. The colors of the circles represent the GO functional groups. The group leading term is the most significant term of the group. The size of the circles reflects the statistical significance of the terms. **c** Hierarchical clustering analysis of the mucosa proteomes based on the correlation matrix between the seven regions: cardia (Ca), fundus (Fu), lesser curvature (LC), greater curvature (GC), angular incisures (AI), antrum (An), and pylorus (Py). The region-based stomach mucosa samples are grouped into two sections: the proximal and distal sections of the stomach. **d** Principal component analysis (PCA) of the 82 mucosa samples based on protein profiles. **e** Volcano plot displaying the differentially expressed proteins in the proximal and distal sections by applying a fourfold change expression difference with $p < 0.05$ (Student's $t$ test). Proteins that were significantly enriched in the distal/proximal sections were highlighted with red/blue filled circles. **f** Representative GO terms of the distal section enriched proteins. **g** Representative GO terms of the proximal section enriched proteins

suggests that despite of being separated into seven regions anatomically, the human stomach can be described as two sections based on the proteomic profiles.

With the above classification, we next searched for section-specific proteins that had fourfold expression difference with statistical significance ($p < 0.05$, Student's $t$ test) between the two substomach sections. Totally, 168 and 176 proteins that meet the above criteria are specifically enriched in the proximal and distal sections, respectively (Fig. 2e, sheet 2 in Supplementary Data 3). For example, gastric markers linked to gastric acid functions (GIF, ATP4A, and ATP4B) as well as a pepsin-related gene (PGA3) are highly expressed in the proximal section (Fig. 2e). In contrast, the growth-related genes, including VGF[34–36] and CDK[37,38] have higher expressions in the distal section (Fig. 2e). Gene Ontology (GO)-enrichment analyses of the proximal section-specific proteins define their roles in the metabolic processes (organic acid, amino acid, carboxylic acid, and so on), while proteins highly expressed in the distal section are enriched in functions in the cell cycle, gland development, regeneration, and DNA/RNA metabolic processes (Fig. 2f, g). The section-specific mucosa proteins seemed to suggest that the proximal section secrets gastric juices to boost the digestion, whereas the distal section functions in cell cycle.

**Region-specific proteins are linked to functional difference.** The co-expression analysis of 2044 high abundance proteins (median iFOT > 1 in at least 1 region, Supplementary Fig. 3c) resulted in 5 protein modules that may suggest distinct functions of the stomach mucosa. For example, module 2 and module 5 seemed to be connected to 2 types of gastric glands (the pyloric glands and the fundic glands) (Fig. 3a, Supplementary Data 4).

The pyloric glands are located in the distal section (Py, An, and AI) where majority of the cells are mucosa secreting cells and G cells that secrete gastrin (GAST). Our proteomic data showed that MUC5AC and MUC6, markers of mucosa secreting cells, and GAST, a marker for G cells, were preferentially expressed in the distal section (Py, An, and Al) (Fig. 3b, Supplementary Fig. 3d). The parietal cells and the chief cells are mainly found in the fundic glands that are located in the proximal section (module 5 in Fig. 3a, Fig. 3c). The proteomics data showed that markers of parietal cells, such as ATP4A and ATP4B, were enriched in the proximal section (Fig. 3b and Supplementary Fig. 3e). Gastrin is known to work in conjunction with parietal cells, leading to secretion of the gastric intrinsic factor (GIF) and hydrochloric acid. The gastric acid further activates the release of pepsinogen (PGA3) as well as gastric lipase (LIPF) from the gastric chief cells. Our data showed that both PGA3 and LIPF were enriched in the proximal section as expected. Hematoxylin–eosin (H&E) staining indicated that both the parietal cells and the chief cells were present in the GC and LC region, while chief cells hardly existed

in the AI region (Fig. 3c). Interestingly, PGC, another well-known gastric chief cell marker, showed no obvious regional difference in the proteomics dataset, suggesting that the regulation of this protein may be more complicated than previously thought (Fig. 3b).

Among the 7 regions, the AI region is unique. The protein co-expression patterns of modules 2 and 3 are very similar, except that a set of proteins in the AI region is expressed at lower levels in module 3 (modules 2 and 3 in Fig. 3a). Both module 2 and 3 contain a large number of co-expressed proteins (287 and 215 proteins, respectively) which are annotated with functions in RNA metabolic process, transcription, and protein localization, among others. Many ribosomal proteins (32/107, module 4 in Supplementary Data 4) are found in the module 4, which are associated with protein translation and secretion of protein in the metabolic processes (module 4 in Fig. 3a). The AI region is the only place where the ribosomal proteins are expressed at low levels. With these observations, we postulate that cell regeneration and protein secretion functions in the AI region may be slower than in the other six regions. This can be due to the fact that the AI region is in the gastric transitional zone, where different types of mucosa meet and the AI region has a substantial population of mixed-type glands composed of parietal cells and very few chief cells[39,40] (Fig. 3c). A schematic diagram summarizing region-specific proteins and their functions in the different regions of the stomach is illustrated in Fig. 3d.

**Construction of the mucosa proteome reference range.** One aim of this study is to construct a reference map from living individuals to answer the question that how variable the human mucosa proteome is. A quantitative reference map can serve as a baseline for differentiating pathological conditions such as GCA from normal. With a sufficient number of mucosa tissue samples, we can build a reference interval (RI) with the distribution of mucosa protein abundance, which could be used for anomaly detection if the protein abundance exceeds the upper limit of the RI. For a query sample $s_j$ (e.g., a GCA patient), an outlier is defined as a protein whose expression level is higher than the upper limit of the RI, and the detection of multiple outliers would further increase the statistical confidence for diagnosis of gastric disorders (Fig. 4a).

To this end, we constructed proteomic RIs derived from 82 human stomach mucosa samples. The upper limit of the RI is defined as P90 + 3*(P90 – P10), where P90 and P10 are the 90th percentile and 10th percentile of the protein abundance in the reference map, respectively. We used 82 samples to generate the normal stomach mucosa reference map which was sufficient to evaluate the rough range with statistically acceptable variation (Fig. 4b). The average CV% of the proteins in all 82 samples was close to 95%, suggesting that the variation among the samples was not large.

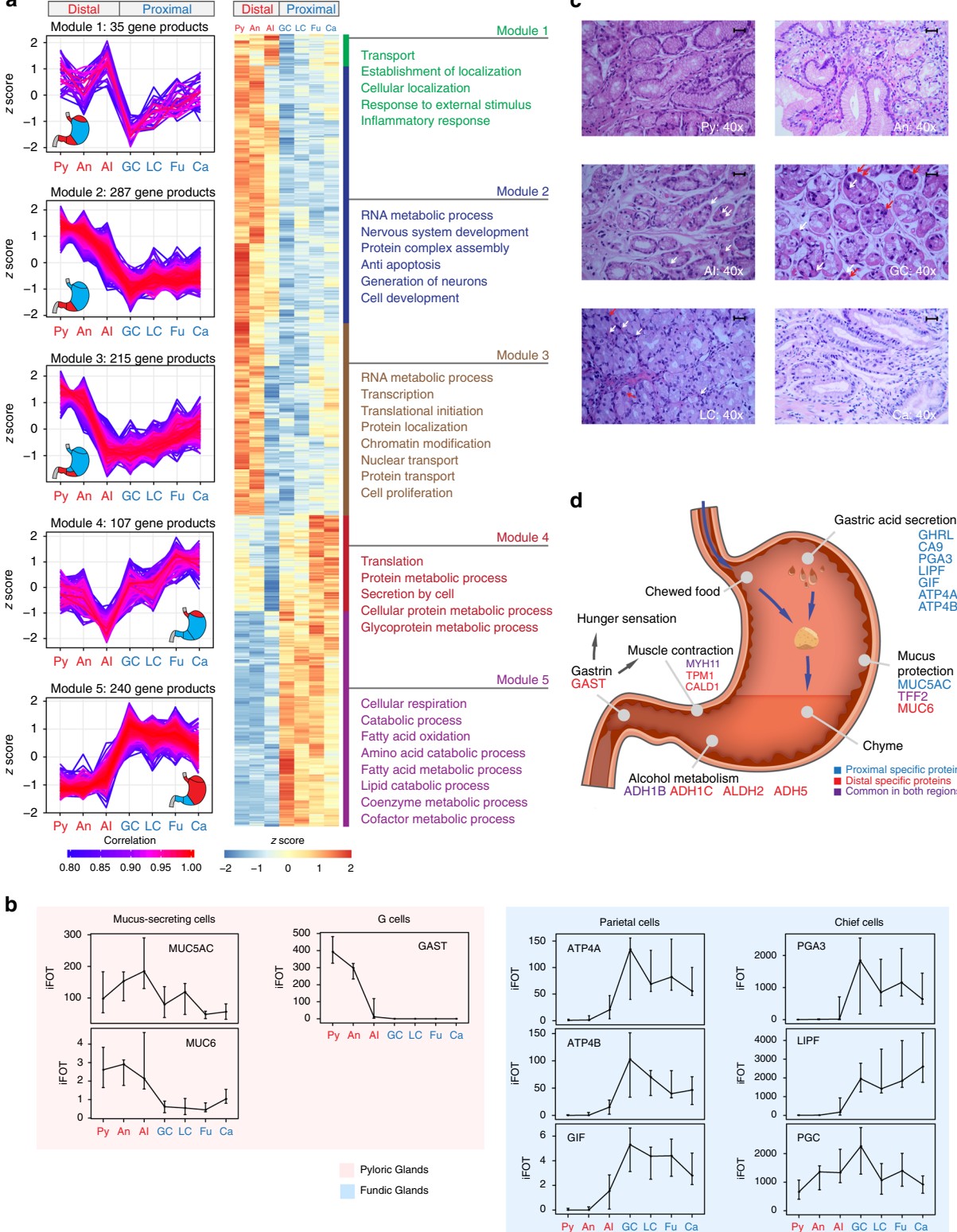

**Fig. 3** Region-specific protein modules and functional differences. **a** Five gene modules revealed by co-expression analysis. Left panel: The co-expression patterns of the proteins in the five module; right panel: representative GO terms of each module. **b** Relative protein abundances of four gastric cell markers across 7 regions (Py: $n = 10$, An: $n = 10$, AI: $n = 15$, GC: $n = 12$, LC: $n = 11$, Fu: $n = 12$, Ca: $n = 12$). Mean ± SD. **c** Hematoxylin–eosin (H&E) staining of mucosa tissues from different stomach regions, indicating the chief cells and parietal cells are mainly found in AI, GC, and LC regions. Red arrows: chief cells; white arrows: parietal cells. Scale bar: 100 μm. **d** A schematic diagram of the structure and functions of the human stomach

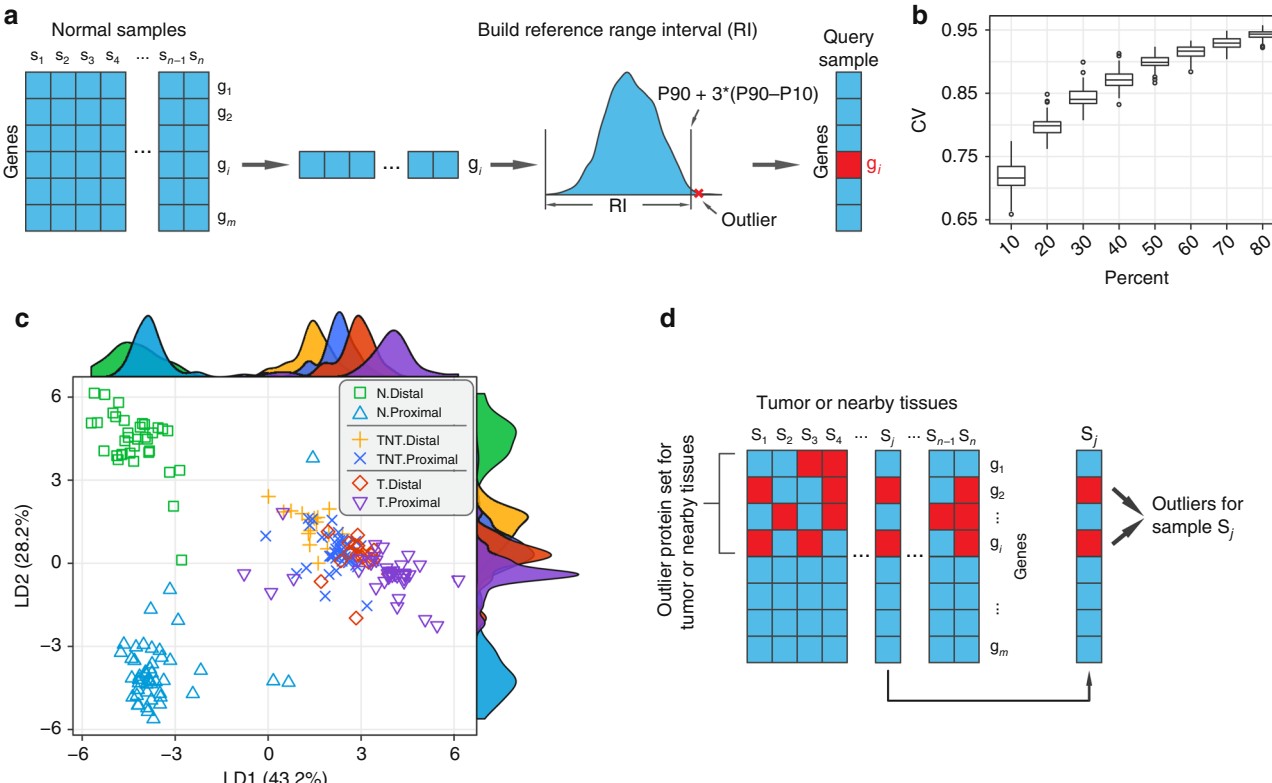

**Fig. 4** The proteome reference range of mucosa. **a** A flowchart to illustrate the construction of the proteome reference intervals (RI) based on 82 normal mucosa samples. Outlier proteins (filled red square) are called if their abundances are out of the upper limit of RI. **b** Boxplot displaying the median abundance of proteins and their variations estimated by subsampling a certain fraction of all samples (10–80%) with replacement (center line: median, bounds of box: 25th and 75th percentiles, and whiskers: from $Q1 - 1.5 \cdot IQR$ to $Q3 + 1.5 \cdot IQR$). **c** Linear discriminant analysis (LDA) of mucosa samples classified into six groups. Green squares: normal tissues from the distal section; sky blue triangles: normal tissues from the proximal section; range pluses: tumor nearby tissues from the distal section; blue crosses: tumor nearby tissues from the proximal section; red diamonds: tumor tissues from the distal section; and violet triangles point down: tumor tissues from the proximal section. **d** A flowchart to illustrate the identifications of anomaly gastric conditions based on the outlier protein sets

As a proof of concept to illustrate the usage of the RI, we next analyzed 58 sample pairs from patients diagnosed with late-stage GCA. We collected T and TNT samples from the patients by endoscope biopsy. Majority of them were diagnosed at the ages of 50–80 and in the advanced stages of GCA in 3 histological subtypes (DGC, IGC, and MGC) (Supplementary Fig. 4a–c). These tissue samples were collected from all seven stomach regions including Ca ($n = 10$), Fu ($n = 8$), LC ($n = 11$), GC ($n = 12$), AI ($n = 6$), An ($n = 10$), and Py ($n = 1$) (Supplementary Fig. 4d, Supplementary Data 5). We mapped the proteomes with the same workflow used in the normal samples and detected 1614–5428 high-confidence GPs in one of the tissues, which yielded a total of 8636 GPs in the T samples and 8245 GPs in the TNT samples (Supplementary Fig. 4e, sheet 2 in Supplementary Data 6).

To test whether the 82 samples in physiological conditions as a whole could be used as the reference for outlier detection, a linear discriminant analysis (LDA) was applied to find linear combinations of variables that may best explain the difference between the six classes of data (normal distal, $n = 35$; normal proximal, $n = 47$; TNT distal, $n = 17$; TNT proximal, $n = 41$; tumor distal, $n = 17$; and tumor proximal, $n = 41$). All data were projected onto the first two linear discriminants, two of which explained 71.4% of the variance within all groups (Fig. 4c). Apparently, the first linear discriminant reflected the pathological status and the second one reflected the location difference of the stomach (i.e., distal vs. proximal). While a clear boundary between the two sections

(proximal and distal) could be seen in the normal samples (N), it became ambiguous in the TNT samples and it disappeared completely in the tumor samples. The results showed that as the carcinogenesis progress, tumors lose their regional characteristics. Moreover, the LD1 appeared to be much more important than the LD2 (variance explained: 43.2% vs. 28.2%) and could clearly separate the three types of samples. With these observations, we conclude that a pan RI built upon the normal tissues from the whole stomach can be used for the detection of tumor outliers.

**Class-specific outlier proteins and GCA subtyping**. We searched the outlier proteins in the N, TNT, and T by using the pan RI (Fig. 4d). The tumor tissues had the most outliers and the normal tissues had the least (Supplementary Fig. 4f). Surprisingly, a substantial number of outlier proteins were identified in the TNT samples. In the present work, a median number of 66, 353, and 435 outlier proteins in the N, TNT, and T samples were called, respectively (Supplementary Fig. 4f, sheet 3 in Supplementary Data 6).

Cancer is a disease that is genetically heterogeneous. The identification of new subtypes can greatly facilitate cancer care for personalized treatment. Based on the strict T outlier protein set (582 GPs) derived from the T samples, three GCA subtypes (T1–3) (Fig. 5a) were found with clear boundaries: T1 had a high fraction of cell cycle related proteins; T2 was enriched with proteins involved in the immune functions; and T3 had metabolic

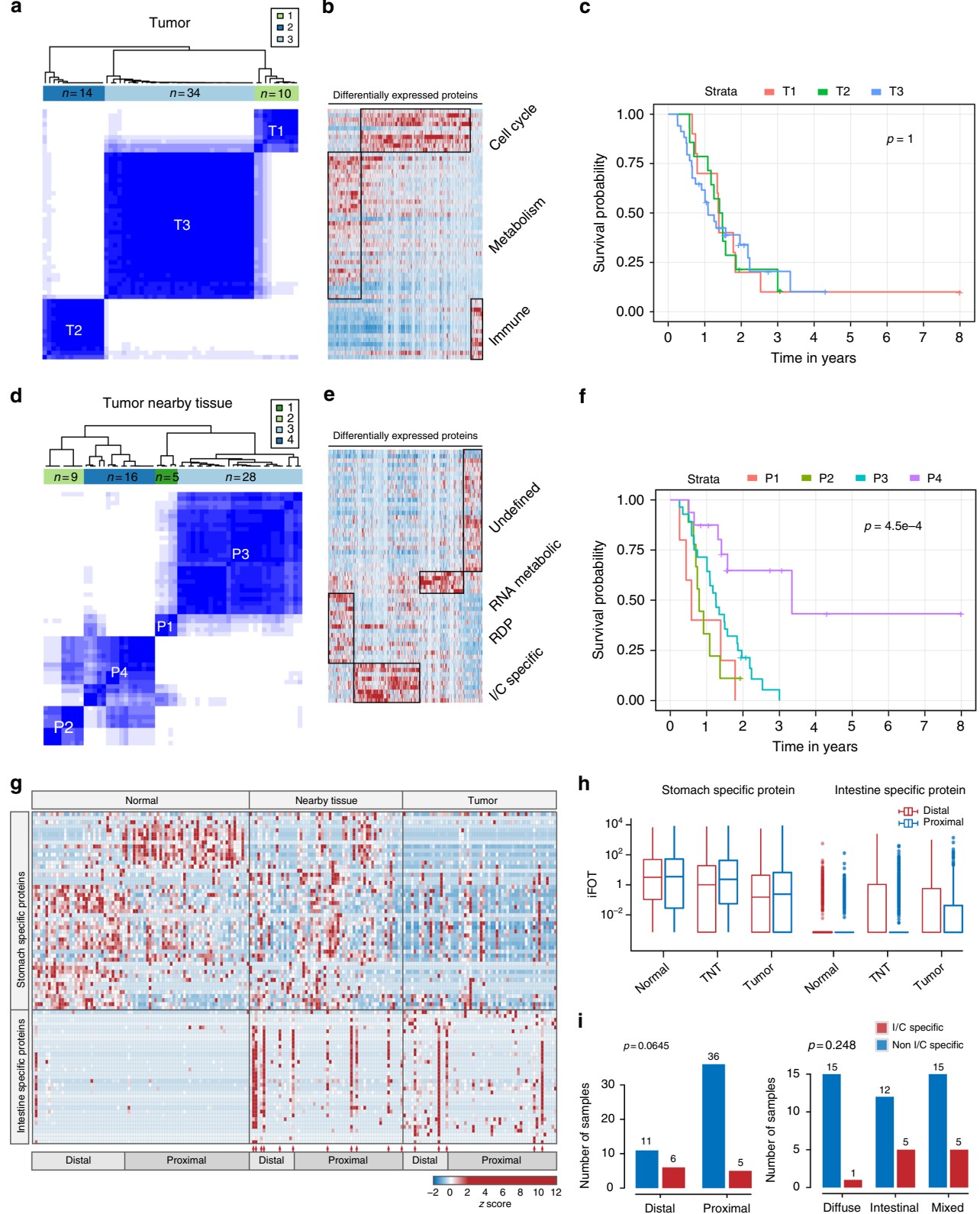

features (Fig. 5b, Supplementary Fig. 5a, Supplementary Data 7). However, these three subtypes are not correlated with the overall survivals (OS) (Fig. 5c).

Applying the same clustering analysis to the TNT outlier proteins identified four subtypes (P1–4) (Fig. 5d): P1 was enriched with the RNA metabolic process; P2 was enriched with proteins that are intestine/colon enriched (I/C enriched); P3,

which contained the largest number of proteins, had no significant functional enrichment; and P4, was enriched with regulation of developmental process (Fig. 5e, Supplementary Fig. 5b, Supplementary Data 7). Interestingly, subtypes determined by TNT outlier proteins are significantly associated with the OS. P2 and P1 subtype is associated with the worst OS, while P4 is associated with the best outcome (Wilcoxon test, $p =$

**Fig. 5** Gastric cancer subtyping by tumor tissues or tumor nearby tissues. **a** The consensus clustering analysis of gastric cancer samples ($n = 58$) based on 582 outlier proteins in tumor tissues. **b** Heatmap display of three GCA subtypes clustered by tumor tissues (T1–T3) and molecular functions of these subtypes. **c** Survival analysis of gastric cancer patients in the three subtypes (T1–T3). **d** The consensus clustering analysis of gastric cancer samples ($n = 58$) based on 377 outlier proteins in tumor nearby tissues. **e** Heatmap display of four GCA subtypes clustered by tumor nearby tissues (P1–P4) and molecular functions of these subtypes. **f** Survival analysis of gastric cancer patients in the four subtypes (P1–P4). **g** Heatmap display of stomach and intestine enriched proteins in the proximal and distal regions respectively and their distributions in the N, TNT, and T samples. The I/C-enriched type samples were indicated by red arrows. **h** The relative abundances of the stomach and intestine enriched proteins in N, TNT, and T, separated by the distal and proximal sections (center line: median, bounds of box: 25th and 75th percentiles, and whiskers: from $Q1 - 1.5*IQR$ to $Q3 + 1.5*IQR$). **i** Bar chart displaying the number of I/C enriched type and non-I/C enriched type samples in the distal and proximal sections (left panel) and three Lauren subtypes (diffuse, intestinal and mixed; right panel)

0.00045, Fig. 5f). Our findings suggest that the proteomic characteristics of TNT instead of tumors themselves may be a better tool than that of T for late-stage cancer subtyping.

The striking feature of the intestine/colon enriched (I/C enriched) proteins in the P2 subtype prompted us to investigate the presence of these proteins in all samples (N, TNT, and T). A list of tissue enriched genes (stomach or intestine enriched) was chosen based on the RNA-seq results from the HPA (The Human Protein Atlas) database (Fig. 5g). We found that the stomach enriched proteins ($n = 49$, Supplementary Data 8), including GIF, ATP4A, and ATP4B, were substantially decreased in TNT and eventually almost disappeared in T (Fig. 5h, Supplementary Data 8), suggesting the loss of the stomach identity in the GCA. The I/C enriched proteins ($n = 35$, Supplementary Data 8) were significantly increased in the TNT; this also occurred to the T but in relatively fewer cases. The H&E staining of the I/C protein enriched cancer subtype shows different cellular morphology than that of typical intestinal metaplasia (IM) (Supplementary Fig. 5c). While the IM samples contained many goblet looking cells revealed by the H&E staining, such cells were not seen in the three I/C subtype TNT samples that we randomly chosen. Moreover, the I/C-enriched subtype seemed to be more prevalent in the distal section than that in the proximal section (Fisher's exact test, $p = 0.0645$) and appeared to be less common in the diffuse-type GCA according to the Lauren classification (Fig. 5i).

## Discussion

In this study, we employed a streamlined proteomics procedure to investigate the gastric endoscopic biopsy samples collected from seven anatomic regions of the stomach. These data allowed us to build a region-resolved reference map that can be used for better understanding of stomach functions and diseases. Importantly, the proteomics data are based on biopsy samples that come from living human beings, not autopsy samples, providing a human mucosa proteomes under near physiological conditions. Our datasets provide a navigation tool to connect the proteome expression to the cellular morphology and anatomy of the stomach mucosa. They could also be useful for the characterization of unknown proteins as well as positioning gastric proteins into their corresponding anatomically resolved regions in the human stomach.

This reference proteome map allowed us to group the seven anatomic regions of the stomach into two sections based on protein expression profiles. The proximal section is associated with gastric juice secretion and the formation of the acidic digestive environment, while the distal section functions in digestion, contraction, and the construction of the mucous barrier to avoid self-digestion. The protein co-expression analysis identified protein modules distributed in different regions. The GO-term enrichment analyses of the modules suggested their potential functions and may predict the role of these proteins whose functions were not known in the stomach.

A goal of this study is to generate a proteome reference map that would provide protein-expression range that covers individual variation and could serve as the baseline to distinguish disease from normal[41–43]. To this end, we utilized 82 samples from the 7 stomach regions to determine the 1.0 version of the expression range of the stomach mucosa proteome. We found that most of the proteins have limited variations among the biopsy samples, suggesting that individual differences are not as great as previously anticipated and permit us to use this reference range as a normal baseline to evaluate abnormal or disease samples to find outlier proteins that deviate from the normal values.

TNTs have been commonly used as normal controls for surgery and to find dysregulated proteins and pathways in tumors. Using this reference map, we found that outlier proteins are present in both tumors and the TNTs. Our analysis clearly shows that TNTs are very different from normal and they may represent a state in which transition to tumor is occurring. Recently, RNA-seq data from breast tissues indicated that TNTs are different from normal[44]. Our data provide proteomic evidence for this notion in GCAs. Traditionally, tumor tissues are used for tumor subtyping in a variety of omics-based discovery studies. It was somewhat unexpected to find that GCA subtypes by the TNT outlier proteins had better correlation with clinical outcomes than those by tumor outliers. This provides another evidence to support the view that TNTs are not equivalent to normal tissues in cancer analysis, and their proteomic characteristics allows for better cancer subtyping. It remains to be investigated whether TNT proteomics analysis can help predict cancer re-occurrence and guide treatment.

The identification of the intestine/colon protein enriched subtype is intriguing in that it is morphologically distinct from the IM and is associated with poor OS. Interestingly, the I/C-enriched protein expression appear to be more prevalent in TNT than in the tumor. It remained to be investigated whether the I/C-enriched cancer subtype represents a molecular cancer subtype and what roles it plays in the initiation and progression of the GCA.

In summary, the region-resolved mucosa proteome provides a physiological reference map that may be useful in clinical applications. Interrogation of this reference map permitted a comprehensive comparison of normal tissue, tumor nearby tissue, and tumor tissue from the human stomach and could provide insights in our understanding of the pathological features of many stomach diseases.

## Methods

**Biospecimen collection**. Human stomach normal mucosa tissues were collected from patients with some mild gastric disorders. Tumor mucosa tissues and the matching nearby mucosa tissues were collected from GCA patients who did not receive chemotherapy. Samples were obtained from the seven anatomic regions of stomach by endoscopic biopsy. They were cleaned with sterile towel, immediately transferred into sterile freezing vials and immersed in liquid nitrogen, then stored at $-80$ °C until use. All cases were histologically evaluated according to the seventh

edition of American Joint Committee on Cancer (AJCC) staging system[45]. Informed consent was obtained from all human participants and collection of human tissue samples was approved by the local ethics committee (KY-2014-6-3) at the Affiliated Hospital, Academy of Military Medical Sciences, Beijing, China.

**Protein extraction, trypsin digestion and LC–MS/MS analysis**. Tissues (82 apparently normal, 58 TNT and 58 tumor tissues with biological and technical replicates: $n = 1$) were minced in buffer (50 mM $NH_4HCO_3$) followed by freezing and thaw 5 min in liquid nitrogen and 5 min in 95 °C heating). The lysate was reduced with 10 mM dithiothreitol (DTT) at 56 °C for 30 min and alkylated with 10 mM iodoacetamide at room temperature in the dark for additional 30 min. About 100 µg of protein samples were then digested using trypsin. Tryptic peptides were separated in a homemade reverse-phase C18 column in a pipette tip with nine fractions using a stepwise gradient of increasing acetonitrile (6, 9, 12, 15, 18, 21, 25, 30, and 35%) under basic conditions (pH 10). The nine fractions were combined into six MS samples, dried in a vacuum concentrator (Thermo Scientific), and then analyzed by LC–MS/MS.

Peptide samples were loaded onto a trap column (100 µm × 2 cm, homemade; particle size, 3 µm; pore size, 120 Å; SunChrom, USA), separated by a homemade silica microcolumn (150 µm × 12 cm, particle size, 1.9 µm; pore size, 120 Å; SunChrom, USA) with a gradient of 5–35% mobile phase B (acetonitrile and 0.1% formic acid) at a flow rate of 600 nl/min for 75 min. LC–MS/MS was performed on an Orbitrap Fusion mass spectrometer using an Orbitrap mass analyzer at a mass resolution of 60,000 (Thermo Fisher Scientific, Rockford, IL, USA) coupled with an Easy-nLC 1000 nanoflow LC system using an ion trap analyzer with the AGC target at 5e3 and maximum injection time at 35 ms (Thermo Fisher Scientific), or a Q Exactive HF mass spectrometer using an Orbitrap mass analyzer at a mass resolution of 120,000 (Thermo Fisher Scientific, Rockford, IL, USA) connected to an UltiMate 3000 RSLCnano System using an Orbitrap mass analyzer at a mass resolution of 15,000 (Thermo Fisher Scientific). The MS/MS analysis was performed under a data-dependent mode. One full scan was followed by up to 20 data-dependent MS/MS scans with higher-energy collision dissociation (normalized collision energy of 35%) or collision induced dissociation (normalized collision energy of 27%). Dynamic exclusion time was set with 18 s.

**Data processing and protein quantification**. MS raw files were processed with the Firmiana proteomics workstation[46]. Briefly, raw files were searched against the NCBI human Refseq protein database (released on 04-07-2013, 32,015 entries) in Mascot search engine (version 2.3, Matrix Science Inc.). The mass tolerances were: 20 ppm for precursor and 50 ppm or 0.5 Da for product ions collected either by Q-Exactive HF or Fusion, respectively. The proteolytic cleavage sites are KR. Up to two missed cleavages were allowed. The database searching considered cysteine carbamidomethylation as a fixed modification and N-acetylation, oxidation of methionine as variable modifications. All identified peptides were quantified in Firmiana with peak areas derived from their MS1 intensity. Peptide FDR was adjusted to 1%. For protein level, we kept the proteins that had at least one unique peptide and two high-confidence peptides (mascot ion score > 20). For protein quantification, we used intensity-based label-free quantification, the so called iBAQ algorithm[47]. We then converted iBAQ value to FOT–iBAQ value of each protein divided by the sum of all iBAQ values of all proteins in the sample; the FOT values were then multiplied by $10^5$ to obtain iFOT numbers to make easy visualization of low abundant proteins[48]. The MS proteomics data have been deposited to the ProteomeXchange Consortium via the iProX partner repository[49] with the dataset identifier PXD011821.

**H&E staining**. Stomach mucosa tissues were fixed in formalin, embedded in paraffin, cut into 5 µm sections. After deparaffinization and rehydration, 5 µm longitudinal sections were stained with hematoxylin solution for 5 min followed by 5 dips in 1% acid ethanol (1% HCl in 70% ethanol) and then rinsed in distilled water. They were next stained with eosin solution for 3 min followed by dehydration with graded alcohol and cleaning in xylene. The mounted slides were then examined and photographed using an Olympus BX53 fluorescence microscope (Tokyo, Japan).

**Bioinformatics and statistical analysis**. The region-specific proteomes acquired by the proteome analysis of seven regions in stomach was visualized by a Circular proteome map generated with the software Circos (version 0.67–7). The functionally organized GO term network of 4742 core proteins was calculated by ClueGO (version 2.5.2) in the software Cytoscape (version 3.6.1). Correlation analysis was performed by using the corrplot package in R software (version 0.84). Briefly, the mean value of FOT of each protein was computed to represent the abundance of the protein in each region. Hierarchical clustering analysis and PCA were implemented in R software. The distances between the rows or columns of a normalized data matrix were computed using the Euclidean distance. Go term enrichment analysis of proteins was based annotations in the Gene Set Enrichment Analysis (GSEA) database (v5.2).

**GCA subtyping**. Consensus clustering of GCA patients was implemented by an R package (ConsensusClusterPlus, version 1.38.0) using hierarchical clustering algorithm (number of iterations = 500; Distance measured: 1-Pearson correlation; proportion of items to sample: 90%). Proteins used for subtyping should be detected in at least 20% of GCA patients ($n = 12$) and 75% of them were called as outliers in either the tumor or nearby tissues. With these criteria, outlier proteins in the tumor ($n = 582$) or nearby tissues ($n = 377$) were used for consensus clustering with up to 10 clusters were considered.

**Reporting summary**. Further information on experimental design is available in the Nature Research Reporting Summary linked to this article.

## Data availability

MS raw files and searching output data are deposited into proteomeXchange via the iProX partner repository with the accession number PXD011821. A reporting summary for this article is available as a Supplementary Information file. All other data supporting the findings of this study are available from the corresponding authors on reasonable request.

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

## Acknowledgments

This work was supported by the National International Cooperation Grant (2014DFA33160, 2012DFB30080), the National Program on Key Basic Research Project (973 Program, 2014CBA02000), the National Key R&D Program of China (2017YFA0505102, 2018YFA0507501, and 2018YFA0507503), the National High-tech R&D Program of China (863 program, 2015AA020108), Beijing Natural Science Foundation (Z131100005213003), Beijing Science and Technology Project (Z181100001918020), and a grant from the State Key Laboratory of Proteomics (SKLP-YA201401). We thank Drs. Pumin Zhang and Jianping Jin for helpful discussions in initiating the project.

## Author contributions

Conceptualization: C.D., Y.W., B.Z., J.X., and J.Q.; Experiment: Z.T., S.Y., M.L., T.G., X.L., Lei S., R.J., C.Z., and Y.T.; Data analysis: X.N., Lan S., Z.T., C.D., Q.Z., and W.L.; Manuscript preparation: C.D., C.Z., X.N., and Z.T.; Funding acquisition: B.Z., J.X., and J.Q.; Resources: Z.T., T.S., W.Z., S.Y., R.J., Y.T., and J.X.; Supervision: Y.W., C.D., B.Z., J.X., and J.Q.

## Additional information

**Competing interests:** The authors declare no competing interests.

