## [Peer Review File · Nature Communications]

Reviewers' comments:

Reviewer #1 (Remarks to the Author):

Ni et al present the proteome of the non-malignant stomach mucosa, divided into 7 regions. These analyses defined the “normal” mucosa proteome, which was later compared to malignant tumors and their adjacent tissue. It is an interesting study, which shows the heterogeneity within the normal tissues and how these change with transformation. Surprisingly, the malignant tumor tissues showed lower heterogeneity, and the adjacent tissues already showed major proteomic changes, and are closer to the malignant tissue. Overall, the manuscript is nicely written and presented, and includes several interesting approaches to data analysis. However, I have major doubts about the basic data processing, which require clarification and re-analysis.

1. The proteomic data analysis was performed with Firmiana proteomics workstation, which is a new program, only published last year by the authors. The data processing workflows are not sufficiently explained in the methods, and also the original manuscript of the program does not provide the necessary details that would allow head to head comparison with the more used programs, such as Proteome Discoverer or MaxQuant. More specifically, there is improper description of the FDR, and of the quantification algorithms. Beyond the added information in the methods section, the authors should provide quantitative comparison to MQ or PD (can be provided only to the reviewers).
2. The entire manuscript uses iFOT values, which are normalized values for each sample. The non-normalized data must be provided to enable examination of the quality of the raw data.
3. Protein tables have to include all the quality measures of each protein, including its score, coverage, peptides etc. As indicated above, the data should include the non-normalized values as well.
4. Throughout the manuscript the authors indicate the numbers of identified proteins, but these numbers are misleading as the proteins are not filtered with a protein FDR cutoff (should be 0.01). Filtration only based on the peptide FDR may result in much higher error rates on the protein level. I acknowledge that the analyses were done after additional filtrations (at least in the first dataset of 80 samples), and assume that these result in low protein FDR. However, the authors keep indicating the higher numbers of unfiltered data. This is improper and misleading the less experienced readers. Of note, the additional filtrations lead to a 40% reduction in the number of identified proteins.
5. Figure 2A shows that the proteins with high abundance appear in all regions. This is expected due to the technology, which is biased towards the more abundant peptides/proteins. The interesting question is related to the less abundant proteins, which show tissue specificity. Are there proteins that are reproducibly seen among replicates of the same region, but are exclusive in only few regions?

6. Figure 1C shows the dynamic range of expressed proteins in the various regions. Are these averages per region? Are these proteins filtered to appear in a minimal number of replicates per region?
7. On the biological side, a major question is the protein specificity to the epithelial cells within the tissue. Figure 3C shows that much of the tissue is non-epithelial and includes immune cells and connective tissue. The authors should show the specificity of the expression pattern for selected proteins that represent the major discussed processes.
8. Line 287 refers to the T outlier set, which includes 5144 proteins. It is not clear how this set was defined. Does it include all T outliers from all samples? How many of the sample-specific outliers significant across the tumor samples? Statistical tests should be added to show the significant changes across patients.

Minor comments:

9. Figure 1 indicates 'in-situ digestion', while according to the materials and methods section they performed FASP digestion. Figure should be corrected since the digestion was not performed in situ.
10. The color code in Figure 2B should be indicated.
11. Figure 2C shows very high correlations between all regions. It does not seem to reflect separation into two regions. What is the significance of this figure panel?
12. Figure 2D should indicate all seven regions in the PCA, rather than only the proximal and distal regions.
13. Line 138 states "On average, the mucosa proteome contains xx proteins". The fact that the authors were able to detect this number of proteins doesn't mean that this is the number of expressed proteins. Most likely they don't identify all proteins and therefore they should refer to these numbers as identified proteins.
14. Lines 221-223 associate the ribosomes with tissue regeneration. However, secretory tissues usually express very high levels of ribosomes due to the necessity to produce and secrete proteins, and not due to high proliferation. In agreement, their own results show that proliferation proteins do not cluster with the ribosomal proteins.
15. The methods section indicates that the authors used urea lysis and followed with reduction at a high temperature. This combination leads to high carbamylation rates. This modification should be added to the database search.
16. Figure S3- what determines the box size?

Reviewer #2 (Remarks to the Author):

Ni and colleagues perform a proteomic study of the human stomach, looking both at distinct anatomic regions of the stomach as well as a comparison of normal and neoplastic tissue. They evaluated 80 'normal' mucosa samples from 7 regions of the stomach from biopsy samples (across 36 patients). Furthermore, they compared tumor and nearby mucosa from 54 gastric cancer patients. While this reviewer is not expert in the detailed mechanics and analytics of this class of proteomic technology, this dataset has clear potential to be a useful guide to the broader field. A limit of the sampling is, as they admit, their use of patients with gastritis or helicobacter and thus not truly normal. However, as this study involves sampling from patients where ethical issues are paramount, this limitation is inherent to this class of research.

My biggest request as a reviewer relates to the overall utility of this study, which is to serve as a reference to the field. As such, it is important (really critical) for the authors to prepare supplemental data to facilitate its use by the community. Included in this, when the authors perform different clustering and such through the paper, the identity of the genes (and key coefficients of correlations) should be included in supplemental tables so that interested readers can evaluate individual proteins driving specific groupings.

A few specific points:

- It would be interesting to compare correlations within patient vs. within region from patients who have multiple samples.
- Is there any data on the diagnosis of the normal tissue (e.g. a systematic signal from patients with helicobacter gastritis?)
- In sections 2e and Figure 3, where the authors discuss different cell types, this would be useful to show a few specific markers (e.g. with IHC) to validate some of the associations they are discussing.
- In the discussion of the AI region, do the authors have a citation for this being a 'hot spot' of cancer? Can the authors be sure there is not some bias leading to difference in this region (such as different purity of mucosa with sampling)? Some validation of markers being different with IF or IHC would be of value.
- What are the outlier proteins with the tumor samples. Please show in supplemental table.
- Loss of stomach specific and increase in intestinal markers—isn't this just IM? They need to discuss well established metaplasia thinking rather than stating this as a new finding.

-5d: do these nearby tissue features correlate with stage or other known features e.g. histology

Reviewer #1 (Remarks to the Author):

Ni et al present the proteome of the non-malignant stomach mucosa, divided into 7 regions. These analyses defined the “normal” mucosa proteome, which was later compared to malignant tumors and their adjacent tissue. It is an interesting study, which shows the heterogeneity within the normal tissues and how these change with transformation. Surprisingly, the malignant tumor tissues showed lower heterogeneity, and the adjacent tissues already showed major proteomic changes, and are closer to the malignant tissue. Overall, the manuscript is nicely written and presented, and includes several interesting approaches to data analysis. However, I have major doubts about the basic data processing, which require clarification and re-analysis.

*1. The proteomic data analysis was performed with Firmiana proteomics workstation, which is a new program, only published last year by the authors. The data processing workflows are not sufficiently explained in the methods, and also the original manuscript of the program does not provide the necessary details that would allow head to head comparison with the more used programs, such as Proteome Discoverer or MaxQuant. More specifically, **there is improper description of the FDR, and of the quantification algorithms.** Beyond the added information in the methods section, the authors should provide quantitative comparison to MQ or PD (can be provided only to the reviewers).*

*We did side-by-side comparisons between Firmiana and Maxquant (MQ) and found that Firmiana is comparable with MQ or PD in terms of protein IDs and quantifications (see **RL Fig. 1**).*

We picked 14 samples (2 samples from each of the 7 regions) and applied the MQ and Firmiana for head to head comparison. The data processing and filtering parameters are listed as below:

MQ: global protein FDR 0.01, LFQ intensity was used.

Firmiana: peptide FDR 0.01, unique peptide ≥ 1 , strict peptide ≥ 2 (ion score >20), iFOT (iBAQ based) intensity was used.

A total of 5,988 and 7,026 GPs (gene products) were identified by Firmiana and MQ, respectively, from the 14 samples (a total of 84 MS runs) according to the parameters set above. Overlap between Firmiana and MQ is 5,554, which covers 92.8% of the GPs of Firmiana, indicating that the FDR level of Firmiana is similar to that of MQ (i.e. FDR 0.01) (**RL Fig. 1a**) and Firmiana may be slightly more strict when 1 unique peptide and 2 strict peptides are required for gene product

calling. The pairwise comparison is shown in **RL Fig. 1b,c**.

We also applied PCA analysis for the above samples after z-score normalization. All the 14 samples analyzed by Firmiana (red point) were spotted right next to those by MaxQuant (blue point), further supporting the notion that there is no big difference between the 2 data processing methods (**RL Fig. 1d,e**).

Moreover, 9 gastric functional genes (ATP4A, ATP4B, GIF, PGA3, LIPF, PGC, MUC5AC, MUC6, GAST) were selected and their mean intensity in each region were calculated by the 2 methods. The line charts (**RL Fig.1f**) show that 8 of 9 proteins shared the same changing trend and exhibited strong correlation between the 2 methods (Pearson Correlation Coefficient ≥ 0.79). The only exception is GAST, where there is a sharp decline in the antrum region on the

MQ's intensity level. As the gastrin (GAST) is released by G cells in the pyloric antrum of the stomach, the Firmiana result (red line) seems to be more consistent with the known biology in this case.

In summary, the above analysis suggests that Firmiana is a reliable data processing workflow that is comparable to a frequently used method. In the revised text, we added more data processing details in the method section to help readers understand Firmiana better.

2. The entire manuscript uses *iFOT* values, which are normalized values for each sample. **The non-normalized data must be provided to enable examination of the quality of the raw data.**

The non-normalized data in iBAQ are provided as requested. Please see supplementary table2-sheet3: the non-normalized data of normal tissues and the quality measure information; table6-sheet4: the non-normalized data of gastric cancer tissues and the quality measure information.

3. **Protein tables have to include all the quality measures of each protein, including its score, coverage, peptides etc. As indicated above, the data should include the non-normalized values as well.**

They are provided along with the non-normalized values (please refer to the above response).

4. Throughout the manuscript the authors indicate the numbers of identified

proteins, but these numbers are misleading as **the proteins are not filtered with a protein FDR cutoff (should be 0.01)**. Filtration only based on the peptide FDR may result in much higher error rates on the protein level. I acknowledge that the analyses were done after additional filtrations (at least in the first dataset of 80 samples), and assume that these result in low protein FDR. **However, the authors keep indicating the higher numbers of unfiltered data. This is improper and misleading the less experienced readers.** Of note, the additional filtrations lead to a 40% reduction in the number of identified proteins.

We have deleted those misleading comments in the revision as suggested.

5. Figure 2A shows that the proteins with high abundance appear in all regions. This is expected due to the technology, which is biased towards the more abundant peptides/proteins. The interesting question is related to the less abundant proteins, which show tissue specificity. **Are there proteins that are reproducibly seen among replicates of the same region, but are exclusive in only few regions?**

This is a good point. We calculated the detected frequency of each protein in each of the 7 regions (**Supplementary Table 2-sheet 4**). We selected the GPs that were reproducibly seen ($\geq 75\%$ of times) in 1 or 2 regions, but were rarely seen ($\leq 25\%$ of times) in 3 or more regions to define them as region-specific proteins. Under these conditions, we found that 19 GPs were Py enriched, 14 GPs were Py/An enriched (**Supplementary Table 2-sheet 5**), and others were distal or proximal enriched. Interestingly, the majority of the Py/An enriched GPs (TRIM22, RNASEH2A, GATA6, SCAF11, EXOSC3, POLD2, KPNA2) mainly function in RNA/DNA metabolic processes (**Supplementary Table 2-sheet 6**). The results are added to **Supplementary Table 2** and commented in the revised text.

6. Figure 1C shows the dynamic range of expressed proteins in the various regions. **Are these averages per region? Are these proteins filtered to appear in a minimal number of replicates per region?**

They are medians per region. We only kept those proteins that were identified in at least two replicates per region. We added more descriptions in the revised text to make it clearer.

7. On the biological side, a major question is the protein specificity to the epithelial cells within the tissue. Figure 3C shows that much of the tissue is non-epithelial and includes immune cells and connective tissue. **The authors should show the specificity of the expression pattern for selected proteins that represent the major discussed processes.**

We thank the reviewer for the suggestion. Now we have added detailed descriptions for epithelial proteins in Fig. 3 (Results, 'Region-specific protein modules are linked spatially to functional differences of the mucosa' subsection).

8. *Line 287 refers to the T outlier set, which includes 5144 proteins. It is not clear how this set was defined. Does it include all T outliers from all samples? How many of the sample-specific outliers significant across the tumor samples? Statistical tests should be added to show the significant changes across patients.*

This set is defined by choosing all T outliers from all T samples. The outlier set was intended to find common outliers that appear in multiple patients. So we excluded those sample-specific outlier proteins to avoid the bias during subtyping. We only kept those outlier proteins that were:

- 1) detectable in at least 20% of gastric cancer patients (N=12), and
- 2) 75% of them were called as outliers in the T and TNT, respectively.

With these two criteria, 582 outliers in the T and 377 outliers in the TNT were selected for further consensus clustering.

Minor comments:

9. *Figure 1 indicates 'in-situ digestion', while according to the materials and methods section they performed FASP digestion. Figure should be corrected since the digestion was not performed in situ.*

Corrected as advised.

10. *The color code in Figure 2B should be indicated.*

Added as required.

11. *Figure 2C shows very high correlations between all regions. It does not seem to reflect separation into two regions. What is the significance of this figure panel?*

Although the stomach is separated into 7 regions, it is a volume of contiguous space after all. By correlating all protein expression across regions we found that the seven region proteome share expression similarity from one region to the next. The result indicates that our data has good consistency with the spatial continuity. Furthermore, we added the y-axis (height) in the figure that provides a measurement of the closeness of either individual data points or clusters calculated by the hierarchical clustering. The separation of two clusters is evident, as the height difference between the 2 clusters is much larger than that of within clusters.

12. Figure 2D should indicate all seven regions in the PCA, rather than only the proximal and distal regions.

We made a few changes as suggested. The seven regions are labeled with seven different shapes while the two different colors indicating the distal and proximal sections are retained for easy visualization.

13. Line 138 states "On average, the mucosa proteome contains xx proteins". The fact that the authors were able to detect this number of proteins doesn't mean that this is the number of expressed proteins. Most likely they don't identify all proteins and therefore they should refer to these numbers as identified proteins.

The sentence is now rephrased as: "On average, the mucosa proteome of our dataset contained 5,670 GPs per region".

14. Lines 221-223 associate the ribosomes with tissue regeneration. However, secretory tissues usually express very high levels of ribosomes due to the necessity to produce and secrete proteins, and not due to high proliferation. In agreement, their own results show that proliferation proteins do not cluster with the ribosomal proteins.

Corrected as suggested.

15. The methods section indicates that the authors used urea lysis and followed with reduction at a high temperature. This combination leads to high carbamylation rates. This modification should be added to the database search.

We apology for the mistake we made in the original manuscript. Protein extraction did not use urea lysis. Tissues were minced in buffer (50mM NH_4HCO_3)

followed by freezing and thaw (5min in liquid nitrogen and 5min in 95°C heating). The lysate was reduced with 10 mM dithiothreitol (DTT) at 56°C for 30 min and alkylated with 10 mM iodoacetamide (IAA) at room temperature in the dark for additional 30 min. About 100 µg of protein samples were then digested using trypsin.

The methods have been corrected in the revision.

16. *Figure S3- what determines the box size?*

The area (box size in figure S3b) is proportional to the number of genes appeared in the GO term. We added detailed description in the figure legend in the revision.

Reviewer #2 (Remarks to the Author):

Ni and colleagues perform a proteomic study of the human stomach, looking both at distinct anatomic regions of the stomach as well as a comparison of normal and neoplastic tissue. They evaluated 80 'normal' mucosa samples from 7 regions of the stomach from biopsy samples (across 36 patients). Furthermore, they compared tumor and nearly mucosa from 54 gastric cancer patients. While this reviewer is not expert in the detailed mechanics and analytics of this class of proteomic technology, this dataset has clear potential to be a useful guide to the broader field. A limit of the sampling is, as they admit, their use of patients with gastritis or helicobacter and thus not truly normal. However, as this study involves sampling from patients where ethical issues are paramount, this limitation is inherent to this class of research.

*My biggest request as a reviewer relates to the overall utility of this study, which is to serve as a reference to the field. As such, **it is important (really critical) for the authors to prepare supplemental data to facilitate its use by the community.** Included in this, when the authors perform different clustering and such through the paper, the identity of the genes (and key coefficients of correlations) should be included in supplemental tables so that interested readers can evaluate individual proteins driving specific groupings.*

We thank this reviewer for the comments and agree that supplemental data is critical to facilitate its use by the community. We have added 8 more supplementary tables in the revision.

A few specific points:

-It would be interesting to compare correlations within patient vs. within region from patients who have multiple samples.

As suggested by the reviewer, we selected one patient (Patient.01) who had 5 samples and 3 patients (Patient.02, Patient.03, Patient.04) who had 4 samples for the correlation analysis. There are strong correlations within regions from patients who had multiple samples, but there was no significant difference when compared with the correlation within patients (RL Fig. 2a,b).

We also checked the expressions of the 9 important gastric functional genes (ATP4A, ATP4B, GIF, PGA3, LIPF, PGC, MUC5AC, MUC6, GAST) in Patient.01 (RL Fig. 2c). The first 5 proteins (ATP4A, ATP4B, GIF, PGA3 and LIPF) are enriched in the proximal section, and the last 3 proteins (MUC5AC, MUC6 and GAST) are enriched in the distal section as expected. However, the expressions of PGA3 and ATP4B reduced in the Fu region. Compared with the median expression of these proteins (Fig. 3b), these results further convinced us that a reference interval is highly needed to eliminate the bias.

-Is there any data on the diagnosis of the normal tissue (e.g. a systematic signal from patients with helicobacter gastritis?)

Yes, the diagnoses of the normal tissues are listed in **Supplementary Table 1-sheet 1**, column 'H. Pylori Infection' (*Helicobacter pylori* infection) and 'Diagnosis'. The column 'Diagnosis' includes superficial gastritis, atrophic gastritis, and bile reflux gastritis.

-In sections 2e and Figure 3, where the authors discuss different cell types, this would be useful to show a few specific markers (e.g. with IHC) to validate some of the associations they are discussing.

The different cell types such as chief cells, parietal cells and mucus-secreting cells are generally morphologically distinguishable. We performed hematoxylin and eosin stain (H&E) and immunohistochemical experiments and showed that the results are consistent to those of proteomics data (as shown in the boxplots): 1) the expression of ATP4A (labeled parietal cells) in lesser curvature (LC) region is higher than that in antrum (An) region.

2) the expression of MUC5AC (labeled mucous cells) in antrum (An) region is higher than that in greater curvature (GC) region.

a.

b.

RL Figure 3 HE and IHC of ATP4A and MUC5AC

a) Hematoxylin-eosin (H&E) and immunohistochemistry (IHC) staining of ATP4A in the region An and LC. Scale bar: 100 μm. The boxplots indicate the iFOT intensity of ATP4A in the 7 regions. b) Hematoxylin-eosin (HE) and immunohistochemistry (IHC) staining of MUC5AC in the region An and GC. Scale bar: 100 μm. The boxplot displaying the iFOT intensity of MUC5AC in the 7 regions.

-In the discussion of the AI region, do the authors have a citation for this being a 'hot spot' of cancer?

We were not able to find a citation for the angular incisures (AI) region being a 'hot spot' of cancer, but just as a rule of thumb, both the clinical doctors and pathologists believe that the incidence of cancer in AI region is significantly

higher than that in the other regions. We decided to delete the sentence in the revised version.

Can the authors be sure there is not some bias leading to difference in this region (such as different purity of mucosa with sampling)?

All samples in this paper were collected by the same surgeon who has done this work for over ten years with rich experience in the endoscopic operation. We also followed uniform requirements and criteria for collecting gastric mucosa under endoscope. We only selected relatively mild lesions that were as close as possible to normal gastric mucosa, and as far away as possible to reduce interstitial (inflammatory cells and fibroblasts) proliferation, thus achieving consistency between patients. We believe that there is no bias leading to difference in this region.

Some validation of markers being different with IF or IHC would be of value.

We used MGST2 as an example. In the proteomic data, the abundance of MGST2 in AI region was much higher than that in other regions. As shown in the figure below, immunohistochemical experiments also validated that the expression of MGST2 in AI region is significantly higher than that in the other regions.

RL Figure 4 IHC of MGST2 in the 7 regions

Immunohistochemistry (IHC) staining of MGST2 in the 7 regions. Scale bar: 100 μ m. Boxplot displaying the iFOT intensity of MGST2 in the 7 regions.

-What are the outlier proteins with the tumor samples. Please show in supplemental table.

The outlier proteins are listed in the **Supplementary Table 6-sheet 3**.

-Loss of stomach specific and increase in intestinal markers—isn't this just IM? They need to discuss well established metaplasia thinking rather than stating this as a new finding.

In the **RL Fig. 5**, we compared the hematoxylin and eosin (HE) staining of 3 randomly chosen TNT samples with small intestine-specific protein expression with a typical intestinal metaplasia (IM) sample. It is clear that there are many goblet cells in the IM samples (blue arrow) that are not seen in the TNTs. The results show that the gastric cancer subtype with intestine-specific proteins is not associated with the IM. However, we can't rule out the possibility that it might be early molecular events occurring during carcinogenesis, which requires our next continuous observation and follow-up.

RL Figure 5 Comparison between I/C specific TNT samples and intestinal metaplasia samples

Hematoxylin-eosin (HE) staining of three I/C specific TNT samples and one intestinal metaplasia (IM) sample. Blue arrows: goblet cells. Scale bar: 100 μ m.

*-5d: do these nearby tissue features correlate with stage or other known features
e.g. histology*

These TNT features have no correlation with stage and histology.

REVIEWERS' COMMENTS:

Reviewer #1 (Remarks to the Author):

The authors replied to my main comments and therefore recommend to acceptance of the manuscript.

There are a couple of technical points that still need addressing:

1. All raw files have to be made publicly available through PRIDE or another alternative.
2. There are still peptide and protein measures that are missing from the supplementary Tables (e.g. peptide score and q-value).

Reviewer #2 (Remarks to the Author):

The authors have done a reasonable job addressing issues for publication and explanation of this very useful dataset. The enhanced supplemental Tables are a critical addition to make these data more accessible.

I would recommend a few minor changes to improve understandability and utility of the data in the cancer sphere.

1. In the description of the tumors, histologic classes are given. Addition of EBV and MSI status would be helpful as these can drastically alter the tumor microenvironment. The authors should then check if their cluster with enhanced survival can be explained by higher rates of EBV or MSI, both associated with better survival.
2. The explanation of the two types of tumor clustering performed in Figure 5 need to be improved so that it is more clear how these approaches differ. I recommend clarification in the text and also adding appropriate schematics in Figure 5 so those who are more visually oriented can understand how these clustering approaches differ from each other.

REVIEWERS' COMMENTS:

Reviewer #1 (Remarks to the Author):

The authors replied to my main comments and therefore recommend to acceptance of the manuscript.

There are a couple of technical points that still need addressing:

1. All raw files have to be made publicly available through PRIDE or another alternative.

Response: MS raw files and searching output data are deposited into proteomeXchange with the accession number PXD011821.

2. There are still peptide and protein measures that are missing from the supplementary Tables (e.g. peptide score and q-value).

Response: We have tried to include these peptide information into the supplementary tables. However, the file size would exceed the upload limit (150MB) because of the large number of peptides (N>200,000). Moreover, all the measurement information can be found in the MS searching output data (PXD011821).

Reviewer #2 (Remarks to the Author):

The authors have done a reasonable job addressing issues for publication and explanation of this very useful dataset. The enhanced supplemental Tables are a critical addition to make these data more accessible.

I would recommend a few minor changes to improve understandability and utility of the data in the cancer sphere.

1. In the description of the tumors, histologic classes are given. Addition of EBV and MSI status would be helpful as these can drastically alter the tumor microenvironment. The authors should then check if their cluster with enhanced survival can be explained by higher rates of EBV or MSI, both associated with better survival.

Response: None of the 58 gastric cancer patients received EBV/MSI testing.

2. The explanation of the two types of tumor clustering performed in Figure 5 need to be improved so that it is more clear how these approaches differ. I recommend clarification in the text and also adding appropriate schematics in Figure 5 so those who are more visually oriented can understand how these clustering approaches differ from each other.

Response: We have made some changes to the Figure 5. Please see below.

Figure 5

Each column indicates a differentially expressed protein